# Obesity and recovery from acute kidney injury (Ob AKI): a prospective cohort feasibility study

Helen L MacLaughlin,[1,2] Rochelle M Blacklock,[1,2] Kelly Wright,[3] Gerda Pot,[2] Satish Jayawardene,[3] Christopher W McIntyre,[4] Iain C Macdougall,[3] Nicholas M Selby[5,6]

[1]Department of Nutrition and Dietetics, King's College Hospital NHS Foundation Trust, London, UK
[2]Department of Nutritional Sciences, King's College London, London, UK
[3]Department of Renal Medicine, King's College Hospital NHS Foundation Trust, London, UK
[4]Division of Nephrology, University of Western Ontario, Ontario, UK
[5]Centre for Kidney Research and Innovation, University of Nottingham Faculty of Medicine and Health Sciences, Derby, UK
[6]Department of Renal Medicine, Royal Derby Hospital, Derby, UK

**Correspondence to**
Dr Helen L MacLaughlin;
helen.maclaughlin@nhs.net

## ABSTRACT

**Objectives** To test the methodology of recruitment, retention and data completeness in a prospective cohort recruited after a hospitalised episode of acute kidney injury (AKI), to inform a future prospective cohort study examining the effect of obesity on AKI outcomes.

**Design** Feasibility study.

**Setting** Single centre, multi-site UK tertiary hospital.

**Participants** 101 participants (67M; 34F) with a median age of 64 (IQR 53–73) years, with and without obesity, recruited within 3 months of a hospitalised episode of AKI.

**Outcome measures** Feasibility outcomes were recruitment (>15% meeting inclusion criteria recruited), participant retention at 6 and 12 months (≥80%) and completeness of data collection. Exploratory measures included recovery from AKI (regaining >75% of pre-AKI estimated glomerular filtration rate [eGFR]) at 6 months, development or progression of chronic kidney disease (CKD) (kidney function decrease of ≥25% + rise in CKD category) at 12 months, and associations with poorer kidney outcomes.

**Results** 41% of eligible patients consented to take part, exceeding the target recruitment uptake rate of 15%. Retention was 86% at 6 months and 78% at 12 months; 10 patients died and three commenced dialysis during the study. Data were 90%–100% complete. Median BMI was 27.9 kg/m² (range 18.1 kg/m²–54.3 kg/m²). 50% of the cohort had stage 3 AKI and 49% had pre-existing CKD. 46% of the cohort met the AKI recovery definition at 6 months. At 12 months, 20/51 patients developed CKD (39%) and CKD progression occurred in 11/49 patients (22%). Post-AKI interleukin-6 and cystatin-C were associated with 12 months decline in eGFR.

**Conclusions** Feasibility to conduct a long-term observational study addressing AKI outcomes associated with obesity was demonstrated. A fully powered prospective cohort study to examine the relationships between obesity and outcomes of AKI is warranted.

## INTRODUCTION

Acute kidney injury (AKI) is defined as a sudden reduction in urine output or rise in serum creatinine.[1] Standardisation of the definition of AKI[2] has enabled the severity, risk factors and mortality associated with

### Strengths and limitations

► A prospective cohort study was conducted in a generalisable cohort of patient experiencing acute kidney injury (AKI), with follow-up at 6 and 12 months.
► Both creatinine and cystatin-C were measured during follow-up to estimate kidney function.
► Feasibility outcomes were established prior to study commencement.
► Obesity may increase the bias in kidney function prediction equations, which is a limitation of this study.
► As a feasibility study, the study is not powered to determine the effect of obesity on outcomes after AKI.

AKI to be studied. Hospitals in the National Health Service (NHS) in England and Wales identify episodes of AKI using a standardised algorithm based on serial serum creatinine measures.[3] Around 60% of AKI episodes are stage 1, the least severe, with 20% stage 3, the most severe, and overall in-hospital mortality with AKI is high.[4] Full recovery from AKI was previously thought to be associated with no future disease risk, yet data have emerged from experimental animal studies and human observational cohorts demonstrating an increased likelihood of permanent kidney damage, with an almost nine times greater risk of later development of chronic kidney disease (CKD) after AKI, compared with those without AKI.[5–9]

Previous studies addressing factors influencing post-AKI recovery of kidney function and the future development of consequential CKD have generally been retrospective and utilised administrative datasets.[9] A recent meta-analysis and subsequent large retrospective population cohort study identified pre-existing CKD as a significant modifier of outcomes post-AKI, indicating that future studies should classify patients by not only AKI severity, but also by pre-AKI kidney

function.[10][11] In a prospective, matched cohort study, non-recovery of premorbid kidney function at 90 days post-AKI was identified as a risk factor for longer-term development or progression of CKD, when compared with hospitalised patients without AKI.[8]

Observational studies suggest that obesity is an independent risk factor for the development and progression of CKD[12][13] even after adjustment for the related risk factors hypertension and diabetes.[14–16] Obesity is also associated with an increased risk and greater severity of AKI.[17–19] However, the effect of obesity on kidney outcomes after AKI is not known. Several mechanisms have been described by which obesity may carry an increased risk for longer term kidney damage after AKI. Inflammation and increased intra-abdominal pressure, which may increase AKI risk in obesity, coupled with known obesity-related kidney damage due to glomerulopathy and hyperfiltration,[20] may hamper recovery of kidney function or exacerbate existing damage and contribute to an increased risk of CKD development or progression in patients with obesity experiencing AKI. These observations lead to a plausible hypothesis that obesity may influence the risk of future CKD development or progression after AKI.

There are also some methodological challenges in the longitudinal study of renal function in patients with obesity. Interpreting renal recovery with obesity may be complex, as the four variable Modification of Diet in Renal Disease study equation[21] in widespread use may underestimate measured glomerular filtration rate (GFR).[22] Two recent observational studies suggest that the CKD Epidemiology Collaboration (CKD-EPI) equations[23][24] give closer estimates of measured GFR with obesity.[25][26] The CKD-EPI creatinine equation has been shown to perform best in a group of patients including those with established CKD,[26] while the addition of cystatin-C to the CKD-EPI equation was the closest predictor of measured GFR in obesity with normal kidney function.[25][26] In addition, there may be changes in body composition in the period of recovery after an episode of AKI that could also potentially affect estimating equations based on serum creatinine concentration. Prospective studies are required to adequately address these challenges.

Therefore, prior to future, large cohort studies to establish the effect of obesity on long-term outcomes after an episode of AKI, we performed a feasibility study to test the proposed study design and establish recruitment, retention and data collection rates.

## MATERIALS AND METHODS

A prospective cohort study was conducted from June 2015 to May 2017 in a large, multi-site, tertiary referral university hospital in London, UK, providing all major medical and surgical specialties, including a nephrology tertiary referral service. The study was primarily designed to examine the feasibility of recruiting and retaining patients in a cohort study exploring the relationship between obesity and the development or progression of CKD after an episode of AKI. The aim was to recruit a sample size of 100 patients over 26 weeks. It was anticipated that at least 30% patients would be obese, as obesity accounts for at least one in eight hospital admissions in women in the UK,[27] providing an adequate sample of obese participants in the total sample. This was predicted to be adequate to calculate the rates of recruitment and retention and ease of data collection adequately and efficiently, yet in a large enough sample population to assess feasibility for the full cohort study.

Eligible participants were patients aged 18–85 years who had experienced an AKI at any point during hospitalisation, who also had at least one previous serum creatinine measurement available within the previous 12 months. Baseline kidney function was determined from the most recent stable serum creatinine value available within the last 12 months, using the CKD-EPI creatinine equation to determine estimated glomerular filtration rate (eGFR$_{creat}$).[23] Potential participants were identified by screening reports from an electronic AKI detection system using the NHS algorithm developed with the Kidney Disease Improving Global Outcomes (KDIGO) definition of AKI,[28] or identified through the tertiary nephrology referral service using the full KDIGO definition of AKI, or through manual searches of daily serum creatinine values, prior to the implementation of the electronic AKI detection system. Exclusion criteria were pre-AKI eGFR$_{creat}$ less than 20 mL/min/1.73 m$^2$,[23] inability to provide informed consent, receiving palliative care and high likelihood of death during hospitalisation. A member of the research team confirmed each episode of AKI, and excluded progressive CKD cases without AKI.

Patients were recruited and consented either during or just after their hospital admission with AKI. The baseline study visit was planned to occur within 4 weeks of hospital discharge, and included recording of pre-AKI kidney function, diabetes status, smoking status, medications, age, gender, ethnicity, stage and cause of AKI (if known), measurement of height, weight, waist circumference, and collection of blood and urine samples. Further study visits were completed at 3, 6 and 12 months post-AKI onset, and measurements and sample collections were completed according to the schedule in table 1.

Using a wall-mounted stadiometer, the height was measured to the nearest 1 cm without the patients wearing shoes. With the patient wearing light clothing and no shoes, weight was measured to the nearest 0.1 kg on a calibrated electronic scale. Waist circumference was measured to the nearest 0.1 cm at the level of the umbilicus with a calibrated plastic tape measure. Obesity was defined as a body mass index (BMI=weight in kilograms/height in metres squared)≥30 kg/m$^2$ and categories of BMI were described according to the WHO classification system for obesity.[29] Conicity index was calculated as waist circumference (m)/0.109 x √(weight (kg)/height (m)). Patients who commenced dialysis during the study follow-up period were coded as having progressive

| Table 1 | Schedule of data collection | | | | |
|---|---|---|---|---|---|
| Data collected | Time after AKI | <4 weeks | 3 months | 6 months | 12 months |
| Height | | • | | | |
| Serum leptin, adiponectin | | • | | | |
| Weight, waist circumference | | • | | | • |
| Serum interleukin 6, C reactive protein | | • | | | • |
| Serum creatinine, cystatin-C | | • | • | • | • |
| Urinary protein to creatinine ratio | | • | • | • | • |
| Additional plasma, serum and urine | | • | • | • | • |

AKI, acute kidney injury.

CKD, and no further measures of serum creatinine were undertaken.

Pre-specified success criteria for the feasibility outcomes of the study were: (1) at least 15% of those invited to join the study consent to take part and (2) at least 80% retention at 6 and 12 months. Secondary outcomes included data collection rates and prevalence of AKI across BMI categories. Associations between baseline BMI, waist circumference and conicity index and markers of adiposity, inflammation, proteinuria, and the change in eGFR$_{creat}$ from pre-AKI to 12 months post-AKI were examined using Spearman rank correlation, in patients with, and without pre-existing CKD. Exploratory outcomes included the degree of recovery from AKI up to 6 months post-AKI by baseline BMI, the prevalence of development or progression of CKD 12 months post-AKI, and a comparison of estimates of kidney function using eGFR$_{creat}$ and the CKD-EPI equation for creatinine combined with cystatin-C (eGFR$_{creat+cysC}$).[24] Recovery of kidney function was defined as a percentage of the pre-AKI baseline eGFR$_{creat}$.[23] Full recovery was >75%, partial recovery 50%–75% and non-recovery if requiring dialysis or <50% of pre-AKI eGFR$_{creat}$ was retained at 6 months. At 12 months, progression was defined using the KDIGO criteria of a 25% reduction in eGFR$_{creat}$ together with a decline of at least one category of CKD staging,[2] OR commencement of renal replacement therapy. Risk ratios for recovery at 6 months and development or progression of CKD at 12 months in obesity compared with normal BMI were calculated separately for those with and without pre-existing CKD.

### Patient and public involvement

Patients and public representatives were consulted on the methods of recruitment and the burden of study follow-up visits. As a result of the consultation the recruitment period was extended post discharge to reduce the burden of decision making at the time of acute illness. Follow-up study visits were extended to all hospital sites as a result of the consultation. Patient representatives reviewed and refined the patient information sheet and lay summary of the study design. All patients were informed of the study findings by letter and were invited to attend one of two group sessions to give feedback on how the study was conducted and developed a summary of the study findings.

## RESULTS

### Participants

Recruitment took place from June 2015 to May 2016. During the recruitment period 706 patients were identified as having an episode of AKI, with 242 patients eligible who met the inclusion criteria for the study (figure 1). Major reasons for ineligibility were age, death in hospital or receiving palliative care, inability to attend study visits due to distance from hospital to usual home address (international or patients from out of area referred into specialist tertiary services), and no pre-AKI serum creatinine value within the previous 12 months. Of the 242 eligible patients, 232 patients were invited to participate in the study by the time the study reached full recruitment. One hundred and one patients consented to participate

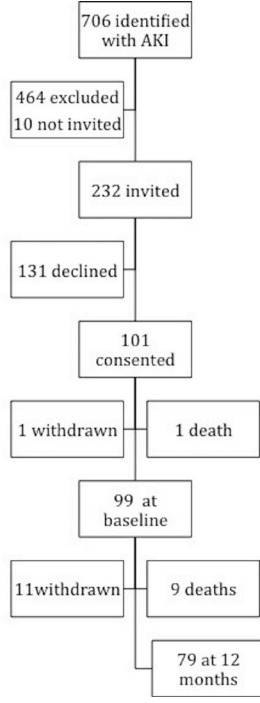

**Figure 1** Recruitment flow diagram. AKI, acute kidney injury.

**Table 2** Baseline characteristics for (median IQR) of patients with AKI in a hospital population. Data are presented as median (IQR) or counts

| | BMI 18 kg/m$^2$–24.9 kg/m$^2$ | BMI 25 kg/m$^2$–29.9 kg/m$^2$ | BMI ≥30 kg/m$^2$ |
|---|---|---|---|
| Participants | 31 | 28 | 40 |
| Age (years) | 69 (50, 74) | 64 (60, 71) | 59 (53, 73) |
| Sex | 21M; 10F | 19M; 9F | 27M; 13F |
| Ethnicity | 22 White<br>9 Black/Asian/Other | 18 White<br>10 Black/Asian/Other | 22 White<br>18 Black/Asian/Other |
| Weight (kg) | 60.6 (55.7, 67) | 80.6 (73.1, 83.9) | 103.2 (94.4, 114.4) |
| BMI (kg/m$^2$) | 21.3 (19.8, 23.5) | 26.7 (25.6, 28.5) | 35.7 (31.9, 38.8) |
| Waist circumference (cm) | 88 (82.5, 92.8) | 103.6 (98.8, 109) | 123 (115.1, 131.5) |
| Conicity Index | 1.32 (1.27, 1.37) | 1.42 (1.35, 1.46) | 1.44 (1.35, 1.49) |
| Existing CKD (n) | 14 | 12 | 22 |
| Hypertension (n) | 16 | 20 | 30 |
| Diabetes (n) | 4 | 15 | 17 |
| eGFR$_{creat}$ (mL/min/1.73 m$^2$) | 61 (43, 78) | 61 (48, 77) | 55 (46, 86) |
| AKI stage | | | |
| 1 | 9 (32%) | 9 (32%) | 10 (36%) |
| 2 | 8 (40%) | 2 (10%) | 10 (50%) |
| 3 | 14 (27%) | 17 (33%) | 20 (40%) |
| Required dialysis | 8/14 | 5/17 | 10/20 |

AKI, acute kidney injury; BMI, body mass index; CKD, chronic kidney disease; eGFR, estimated glomerular filtration rate.

in the study, one withdrew prior to baseline and one did not attend the baseline assessment and died shortly after, leaving 99 patients in study.

Baseline characteristics for the entire cohort, and by BMI category, are displayed in table 2. Forty per cent of the patients with AKI were obese, 29% were overweight and 31% had a normal BMI. Median BMI was 27.9 kg/m$^2$ (range 18.1 kg/m$^2$–54.3 kg/m$^2$). Two patients with a BMI between 18.1 kg/m$^2$ and 18.5 kg/m$^2$ were incorporated into the normal BMI group. Twenty-eight per cent of patients had AKI stage 1, 20% had AKI stage 2% and 52% had AKI stage 3—with 45% of these requiring renal replacement therapy during their hospital admission. Median eGFR$_{creat}$ was 60 mL/min/1.73 m$^2$ and 48% of the cohort had pre-existing CKD stage three or higher prior to entry into the study. Seventy patients had a predominantly likely pre-renal cause for their AKI, 11 patients had a primarily obstructive cause, 12 patients had AKI attributed to the kidneys and in six patients the AKI was deemed to have been multifactorial.

## Recruitment and retention

Screening and recruitment for a target of 100 patients was completed in 12 months; 6 months longer than the planned recruitment time. Electronic identification of AKI was not available for the first 9 months of recruitment so manual searches of electronic data were conducted. Over the 12 months of recruitment, on average 59 patients were screened each month, 20 were eligible for the study and 8.4 patients per month consented to

participate. Overall, 41% of eligible patients approached to join the study consented to take part, exceeding the target recruitment uptake rate of 15%.

By 6 months, a total of 3 patients had died and eight patients withdrew, including one who was too unwell to provide ongoing consent. This left 88 patients in the study, including two requiring dialysis (88/101, 87%), although six did not have a study visit at 6 months (three did not attend, three declined the visit but remained in the study). By 12 months, 10 patients had died, 12 had withdrawn and 79 remained in the study, including three requiring dialysis, giving a final retention rate of 78% (79/101) of consenting patients in the study at 12 months. Eighty-six per cent of patients still living 12 months post-AKI, attended their final study visit.

Data were 100% complete for height, weight, reason for admission, diabetes and hypertension status, blood sampling and AKI staging including identifying the likely cause and classification as pre-renal, renal or post-renal. Urine sampling was 96% complete and waist circumference was measured in 90% of the sample. The median time to the baseline study visits from the date of the peak creatinine was 44 (IQR 32–58) days. Baseline data collection was planned to be completed 1-month post hospital discharge after AKI. However, many patients were either still hospitalised, or not well enough or ready to consider providing informed consent at this point, so the protocol was amended to enable the baseline visit to be completed within 3 months of the AKI episode. Median time to

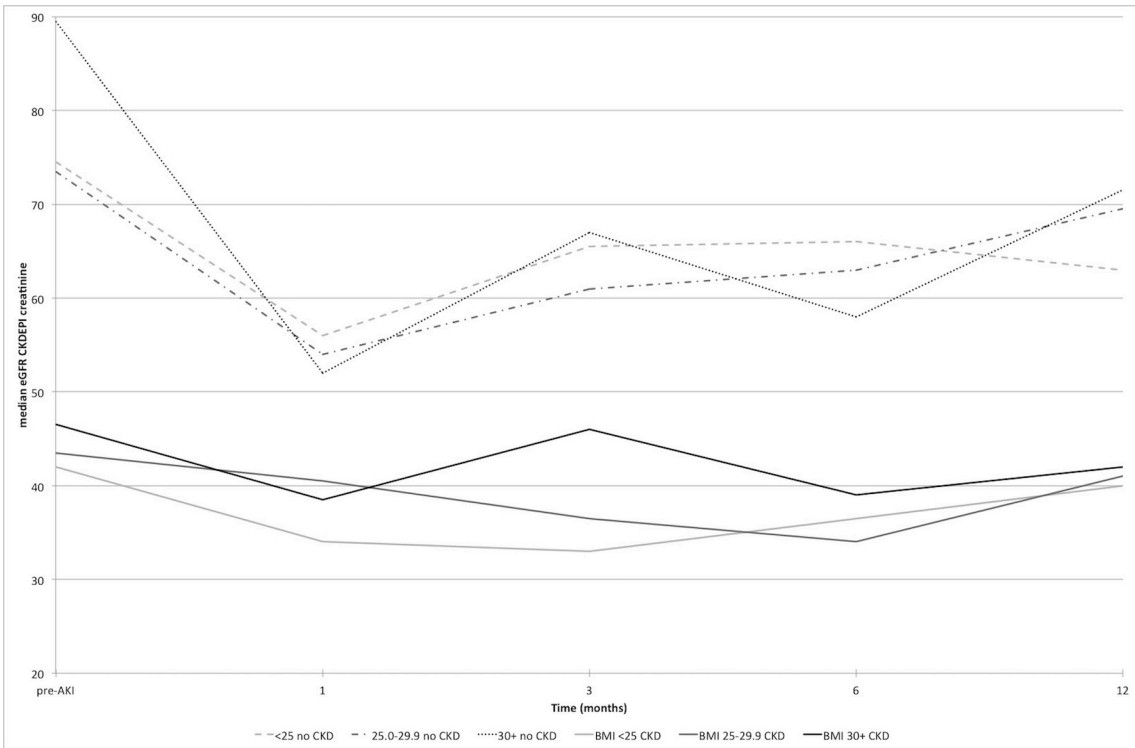

**Figure 2** Estimated glomerular filtration rate (eGFR_creat) from pre-AKI to 12 months post-AKI by BMI category with and without pre-existing CKD. AKI, acute kidney injury; BMI, body mass index; CKD, chronic kidney disease; CKD-EPI, CKD-Epidemiology Collaboration.

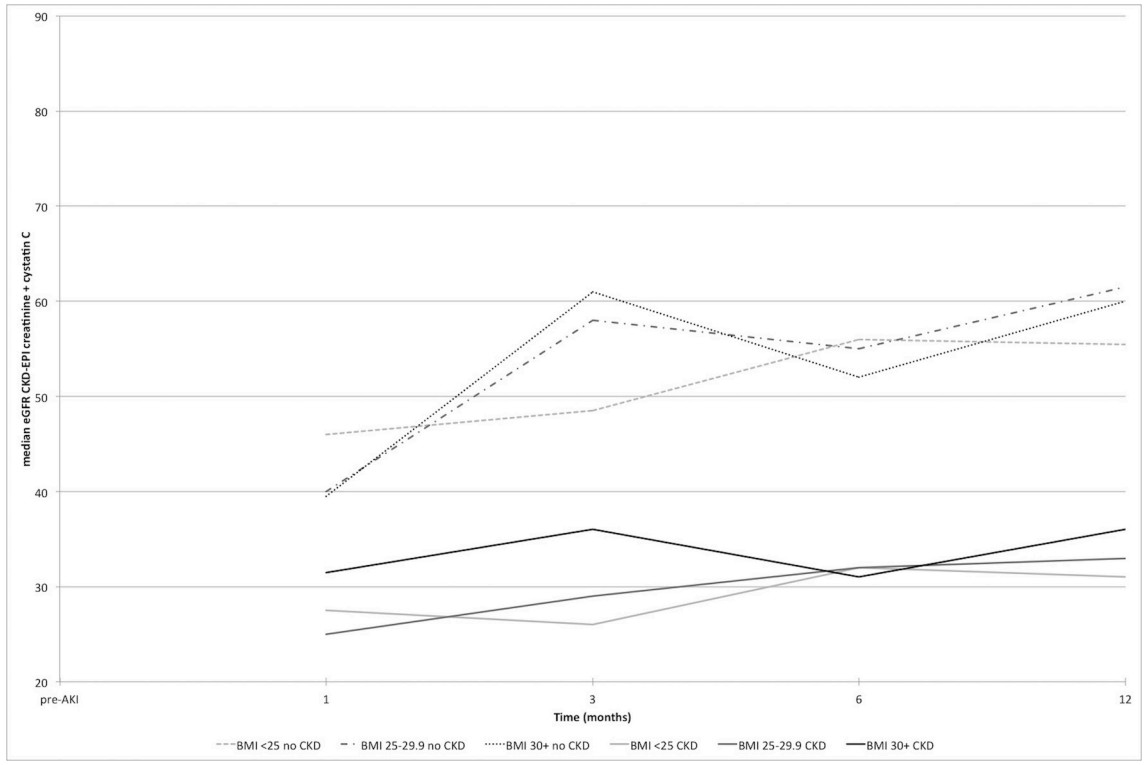

**Figure 3** Estimated glomerular filtration rate (eGFR_creat+cysC) from baseline to 12 months post-AKI by BMI category with and without pre-existing CKD. AKI, acute kidney injury; BMI , body mass index; CKD, chronic kidney disease; CKD-Epidemiology Collaboration.

**Table 3** Incidence rates for kidney function recovery at 6 months and CKD incidence and progression 12 months post acute kidney injury

| | BMI 18 kg/m$^2$–24.9 kg/m$^2$ | | BMI 25 kg/m$^2$–29.9 kg/m$^2$ | | BMI ≥30 kg/m$^2$ | |
|---|---|---|---|---|---|---|
| Recovery >75% eGFR at 6 months | | | | | | |
| No CKD | 47.1% | 8/17 | 69.2% | 9/13 | 35.3% | 6/17 |
| Pre-existing CKD | 47.1% | 8/13 | 50% | 5/10 | 71.4% | 10/14 |
| Progression >25% decline and decrease in CKD stage at 12 months | | | | | | |
| No CKD | 69.2% | 9/13 | 41.7% | 5/12 | 37.5% | 6/16 |
| Pre-existing CKD | 27.3% | 3/11 | 27.3% | 3/11 | 31.3% | 5/16 |

BMI, body mass index; CKD, chronic kidney disease; eGFR, estimated glomerular filtration rate.

6-month and 12-month visits were 186 (180-193) days and 367 (361-373) days, respectively.

### Kidney function and BMI

Estimated kidney function at all study time points is displayed in figure 2, using eGFR$_{creat}$, from pre-AKI to 12 months after AKI, and in figure 3 and with eGFR$_{creat+cysC}$ from baseline to 12 months. There is a clear decline in eGFR, which persists at 12 months post-AKI, particularly in the non-CKD group. With obesity, there is greater undulation in the eGFR recovery pattern compared with the other BMI groups.

The incidence rates for eGFR recovery at 6 months and development or progression of CKD at 12 months for patients are displayed in table 3. At 6 months, 46 patients had recovered >75% of baseline eGFR$_{creat}$, 31 had partial recovery of eGFR$_{creat}$ (50%–75%), seven had not recovered (<50% baseline eGFR$_{creat}$, or dialysis), three patients had died and 14 had no data available. In those with normal kidney function pre-AKI, the relative risk for recovery of baseline kidney function at 6 months with obesity, was not significant (relative risk 0.75; 95% CI 0.3 to 1.7) compared with normal BMI. In patients with pre-existing CKD, there was also no difference in likelihood of recovery to baseline eGFR$_{creat}$ between obesity and normal BMI (relative risk 1.3; 95% CI 0.8 to 2.4).

At 12 months, 10 patients had died, 31 patients had progressive or incident CKD, including two requiring renal replacement therapy. Forty-eight patients had recovered pre-AKI kidney function and 12 had withdrawn from the study. In patients with pre-existing CKD, progression occurred in 11 patients (22%). In those with normal kidney function, 20 patients developed CKD after AKI (39%). Obesity did not significantly impact on the relative risk of CKD development (relative risk=0.5; 95% CI 0.3 to 1.1) or progression (relative risk=1.15; 95% CI 0.3 to 3.8), compared with normal BMI.

Median values for clinical parameters and markers of inflammation at baseline and 12 months are displayed in table 4, stratified by BMI category and pre-AKI CKD status. Generally, there were trends for increasing weight and decreasing inflammation over the 12 months following AKI

in all groups. Proteinuria tended to decrease over time in the non-CKD group and increase in the CKD group.

Associations between baseline BMI and 12 months change in eGFR$_{creat}$ and potential markers of risk of CKD progression are displayed in table 5. Baseline interleukin 6 (IL-6) was inversely associated with change in eGFR$_{creat}$ over 12 months in the non-CKD group, and in the pre-existing CKD group and cystatin-C was inversely associated with CKD progression in the non-CKD group only. No other risk markers, including BMI, were significantly associated with eGFR$_{creat}$ change. Leptin, waist circumference and conicity index were strongly associated with BMI, but not with change in eGFR$_{creat}$.

Bland-Altman plots of the mean versus difference for eGFR$_{creat}$ and eGFR$_{creat+cysC}$ equations, by BMI category, are presented in figures 4–6. eGFR$_{creat}$ was consistently higher than eGFR$_{creat+cysC}$ across all categories of BMI and this effect was more marked at higher eGFR levels.

## DISCUSSION

### Feasibility

We have demonstrated that it is possible to recruit and retain patients in a study of longer-term outcomes associated with obesity after hospitalisation with AKI. Recruitment was well above the target, indicating that patients are willing to participate in studies monitoring kidney function after an episode of AKI. Learning from the workload of this study, we expect that one full time researcher would be required to recruit at the target rate of five patients per week, and complete the study data collection visits.

Retention was just below target, due to patient withdrawal and the competing risk of death. When determining the sample size for a longer-term study on obesity associated outcomes after AKI it will be important to account for loss of follow-up due to the competing risk of death, as one and 5 years mortality post-AKI can be >40% and >60%, respectively, in studies utilising healthcare data.[30]

### Timing of data collection and selection of study variables

The initial study visits 1 month post the peak creatinine was demonstrated not to be feasible, as only one quarter of

**Table 4** Clinical parameters and markers of inflammation at 1–3 months and 12 months after acute kidney injury (AKI) (median and IQR)

| Post-AKI | BMI 18 kg/m²–24.9 kg/m² | | BMI 25 kg/m²–29.9 kg/m² | | BMI ≥30 kg/m² | |
|---|---|---|---|---|---|---|
| | 1–3M | 12M | 1–3M | 12M | 1–3M | 12M |
| **No CKD** | | | | | | |
| Weight (kg) | 62.1 (58.1, 66.7) | 67.5 (59.6, 74.9) | 83.1 (77.1, 85.6) | 92.0 (76.1, 97.3) | 110.9 (97.7, 117.7) | 115.0 (110.8, 124.0) |
| hs-CRP (mg/L) | 9.3 (4.1, 29.6) | 3.2 (1.35, 13.3) | 7.65 (3.05, 10.95) | 5.1 (2.7, 21.6) | 10.6 (6.2, 17.8) | 9.0 (3.9, 16.25) |
| IL-6 (ng/L) | 5.14, (3.2, 11.62) | 1.21 (1.17, 7.44) | 7.89 (3.92, 15.54) | 11.42 (8.52, 26.74) | 10.75 (5.04, 18.47) | 8.63 (3.72, 9.24) |
| PCR (mg/mmol) | 20.0 (8.3, 41.9) | 7.6 (6.1, 16.2) | 11.7 (6.3, 64.8) | 13.5 (5.0, 55.4) | 20.5 (7.1, 63.9) | 10.3 (6.8, 20.4) |
| **Pre-existing CKD** | | | | | | |
| weight (kg) | 57.9 (54.6, 67.0) | 63.2 (58.9, 73.0) | 75.6 (72.5, 81.1) | 77.8 (73.8, 84.4) | 96.4 (90.3, 107.6) | 98.9 (83.0, 116.0) |
| hs-CRP (mg/L) | 3.4 (1.4, 9.6) | 2.4 (1.5, 16.4) | 6.05 (4.4, 20.15) | 4.1 (0.65, 7.45) | 8.15 (3.7, 19.2) | 7.0 (2.8, 14.8) |
| IL-6 (ng/L) | 6.12 (4.51, 10.23) | 4.63 (3.08, 6.62) | 10.55 (7.55, 18.6) | 7.26 (5.28, 10.0) | 6.36 (4.22, 10.02) | 5.64 (3.62, 11.0) |
| PCR (mg/mmol) | 21.5 (11.3, 36.2) | 24.2 (7.5, 86.6) | 28.9 (11.2, 157.4) | 35.8 (17.7, 201.0) | 26.3 (7.8, 75.3) | 33.6 (6.0, 62.8) |

BMI, body mass index; CKD, chronic kidney disease; hs-CRP, high sensitivity C reactive protein; IL-6, interleukin 6; PCR, urinary protein to creatinine ratio.

patients were able to complete the baseline visit within this time frame. As in other prospective studies, performing the baseline study visit 3 months post the AKI episode is recommended.[31 32]

In this feasibility study, data were collected with a view to determining the optimal markers to select for measurement in larger prospective observational studies, and to formulate hypotheses for mechanisms of obesity associated outcomes after AKI. Waist circumference

and conicity index correlated highly with BMI, however neither measure discriminated differently to BMI in associations with change in eGFR$_{creat}$, adipose tissue hormones, proteinuria or markers of inflammation. Single measures of hormonal markers of adiposity post-AKI, such as leptin and adiponectin, may not provide any additional mechanistic understanding over BMI alone. Together, these results indicate that BMI is an adequate classifier of obesity in this population. IL-6 and cystatin-C, as markers

**Table 5** Spearman rank correlations between body mass index or 12 month change in eGFR with baseline measures of adiposity, inflammation and kidney damage

| | Adiponectin | Leptin | IL-6 | hs-CRP | PCR | Waist | Conicity index | BMI | Cystatin-C |
|---|---|---|---|---|---|---|---|---|---|
| **No CKD** | | | | | | | | | |
| BMI | −0.087 | **0.774** | 0.262 | 0.164 | −0.093 | **0.948** | **0.599** | 1 | −0.01 |
| P value | 0.5 | <0.001 | 0.06 | 0.3 | 0.5 | <0.001 | <0.001 | – | 0.9 |
| eGFR change | 0.07 | −0.083 | **−0.361** | −0.152 | −0.138 | 0.065 | −0.026 | 0.031 | **−0.44** |
| P value | 0.7 | 0.6 | 0.02 | 0.4 | 0.4 | 0.7 | 0.8 | 0.8 | <0.01 |
| **Pre-existing CKD** | | | | | | | | | |
| BMI | **−0.310** | **0.666** | −0.062 | 0.22 | 0.015 | **0.921** | **0.396** | 1 | −0.1 |
| P value | 0.03 | <0.001 | 0.7 | 0.1 | 0.9 | <0.001 | <0.01 | – | 0.5 |
| eGFR change | −0.18 | 0.281 | **−0.410** | −0.021 | −0.21 | 0.186 | 0.098 | 0.215 | −0.05 |
| P value | 0.3 | 0.1 | 0.02 | 0.9 | 0.2 | 0.3 | 0.6 | 0.2 | 0.8 |

BMI, body mass index; CKD, chronic kidney disease; eGFR, estimated glomerular filtration rate; hs-CRP, high sensitivity C reactive protein; IL-6, interleukin 6; PCR, urinary protein to creatinine ratio.

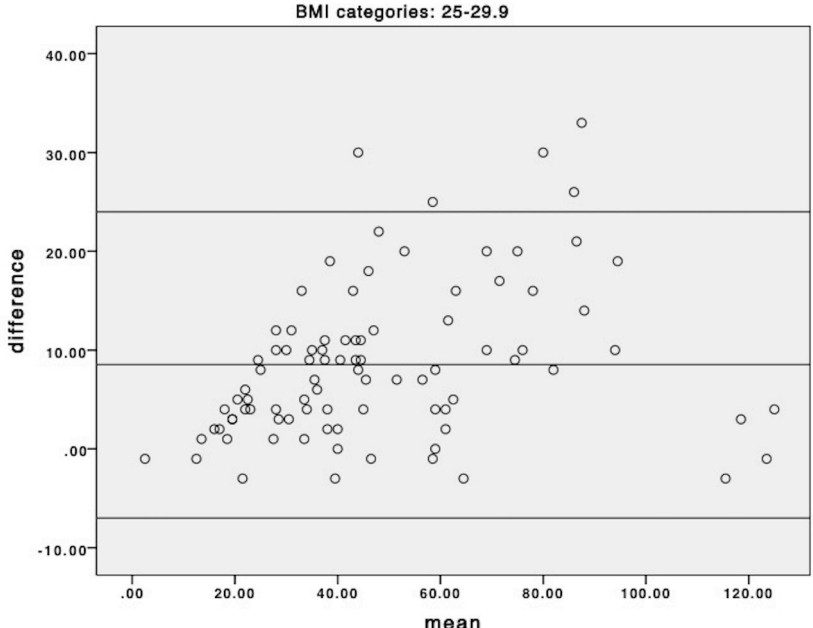

**Figure 4** Bland-Altman plot of mean of individual eGFR$_{creat}$ and eGFR$_{creat+cysC}$ versus difference for the normal weight group (body mass index [BMI] <25 kg/m$^2$); horizontal lines represent mean difference and 95% limits of agreement.

of inflammation and kidney damage, respectively, are likely to provide insight into prediction of risk for CKD outcomes after AKI, as our feasibility data suggest that IL-6 is a more sensitive than high sensitivity-CRP.

### Measurement of eGFR in obesity
The utility of prediction equations for eGFR in obesity remains unclear, particularly when body size or body composition is in flux.[25 33 34] Estimated GFR$_{creat+cysC}$ provided the closest estimate to measured GFR in obesity pre and post bariatric surgery,[25] but has not been studied extensively post-AKI. We chose to examine it as a potential measure of kidney function in this feasibility study. There appeared to be more undulation in eGFR with both equations in patients with obesity, regardless of underlying level of kidney function. This may be attributed to the

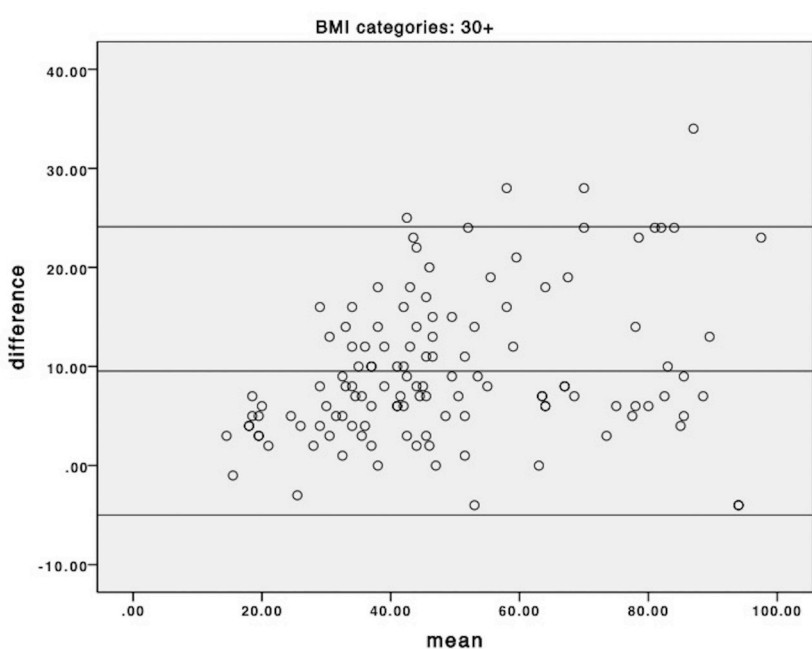

**Figure 5** Bland-Altman plot of mean of individual eGFR$_{creat}$ and eGFR$_{creat+cysC}$ versus difference for the overweight weight group (body mass index [BMI] 25 kg/m$^2$–29.9 kg/m$^2$); horizontal lines represent mean difference and 95% limits of agreement.

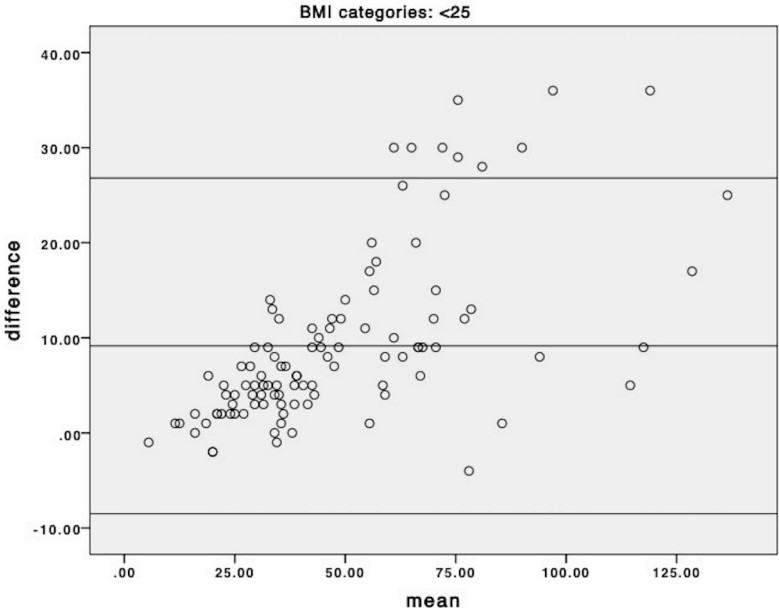

**Figure 6** Bland-Altman plot of mean of individual eGFR$_{creat}$ and eGFR$_{creat+cysC}$ versus difference for the obese group (body mass index [BMI] 30+ kg/m$^2$); horizontal lines represent mean difference and 95% limits of agreement.

known reduced accuracy of eGFR predictions equations with obesity,[26] and highlights the importance of using multiple measures of eGFR over time when examining long-term change in kidney function. Bland Altman plots did not indicate any difference in the bias between the two methods across categories of BMI but did suggest that at higher eGFR levels the estimating equation using cystatin generated lower results than the creatinine only equation. Additionally, there was no association between baseline BMI and cystatin-C in this cohort, and together these may indicate that cystatin-C was not influenced by obesity in this study. Further studies comparing different estimating equations to measured GFR in post-AKI populations are required to determine which methods are valid.

At all-time points post-AKI, eGFR$_{creat}$ was higher, than eGFR$_{creat+cysC}$. Other studies have also found differences in creatinine and cystatin-C based equations in specific populations. In an elderly community population, smoking, age, BMI, inflammation (measured by C reactive protein) and glucocorticoid were associated with higher eGFR$_{creat}$ and lower eGFR$_{cysC}$.[35] The authors suggested that these differences could be due to a reduction in muscle mass and inflammation, respectively.[35] These non-GFR determinants of creatinine and cystatin-C are both also likely evident after AKI and may explain the lower eGFRs observed in the current study with the inclusion of cystatin-C. Cystatin-C correlated with indices of adiposity, although in a previous study the difference between measured GFR and eGFR$_{cysC}$ did not correlate with body fat measured by Dual-energy X-ray absorptiometry (DEXA) in young males with normal kidney function.[36] In that study, eGFR$_{creat+cysC}$ performed best for both

high lean mass and high fat mass groups, although the eGFR$_{creat}$ had greater accuracy for the normal BMI group. Together, these findings suggest that both body composition and inflammation are potential confounding factors for interpretation of estimates of kidney function post-AKI, due to their non-GFR influences on creatinine and cystatin-C. Studies examining eGFR$_{creat+cysC}$ at 90 days post-AKI as a risk factor for development or progression of CKD are warranted.

### Development and progression of CKD

Almost one-third of patients showed development or progression of CKD 1-year post-AKI. Horne and colleagues reported CKD development or progression in 24% of their prospective cohort at 3 years post-AKI,[8] while in those with complete recovery post-AKI, development of stage 3 CKD occurred in 15% of patients after median follow-up of 2.5 years.[5] Fifty per cent of the present cohort experienced stage 3 AKI, while most observational cohorts typically include 60%–70% of patients with stage 1 AKI,.[8 37] Almost half the cohort had existing CKD and more patients required renal replacement therapy during hospitalisation than in the other UK studies,[38] both of which are likely contributors to the high rate of CKD development and progression in the present study.

Raised IL-6 may be a risk factor associated with worsening of eGFR after AKI. One potential contributing factor could be a latent effect of obesity on inflammation, which may influence CKD development after AKI. Whether there is an interaction or 'cross-talk' between obesity and markers of inflammation in CKD progression remains to be studied further in a suitably powered study with longer-term follow-up. Low BMI may also be

associated with inflammation, so future studies should seek to determine if non-linear relationships are evident. This study was not designed to examine if the risk of CKD development or progression is higher with obesity, so the reported lack of difference between obese and non-obese groups could represent type II error. In addition to small sample size, we also recognise the limitation of short-term follow-up of 1 year, and it is possible that risks and outcomes associated with obesity will be more pronounced over longer periods of time. In any fully powered study of CKD outcomes following AKI, other factors such as the severity and duration of AKI, previous or repeated AKI episodes, proteinuria, severity of the illness precipitating the AKI and subsequent interventions, frailty and pre-existing CKD are all likely confounders to be considered when examining the risk of later CKD development or progression.

The tools available to identify AKI limit all studies on outcomes following AKI. While the KDIGO definition of AKI refined previous criteria, it is still limited by changes in creatinine and urine output, both deemed late markers of kidney damage.[28] Furthermore, awareness of AKI risk may influence the frequency of measurement of serum creatinine, the use of an AKI identification algorithm which excludes urinary output identification criteria, and inclusion in studies such as this of only participants with established baseline kidney function, introduce bias in participant identification.[30]

In conclusion, we have demonstrated that it is feasible to conduct a prospective study on the effect of obesity on the long-term outcomes after AKI. Our results inform the design of future definitive studies, which should include measures of pre-AKI kidney function, long-term follow-up after the acute illness and recovery phase, and careful consideration of the use of creatinine, cystatin-C, or measured GFR to assess kidney function.

**Acknowledgements** We would like to thank Sarah Mackie, Rebecca Freeman, Richard Hull, Joble Joseph and Gurminder Khamba for their clinical expertise and assistance in identifying potential participants.

**Contributors** HLML, CWM and NMS contributed to the conception and design of the study. HLML, NMS, GP, IM and SJ developed the methodology. HLML, SJ, RMB and KW collected and analysed the data. HLML drafted the work and all authors revised it critically for important intellectual content. All authors approved the final version submitted and agree to be accountable for all aspects of the work.

**Funding** This work was supported by a National Institute of Health Research (NIHR)/Higher Education England (HEE) Clinical Lectureship award (CAT CL-2014-05-005). This paper presents independent research funded by the National Institute for Health Research (NIHR) and Health Education England. The views expressed are those of the authors and not necessarily those of the NHS, the NIHR or the Department of Health.

**Competing interests** None declared.

**Patient consent for publication** Not required.

**Ethics approval** National Research Ethics Service - London Bloomsbury Committee.

**Provenance and peer review** Not commissioned; externally peer reviewed.

**Data sharing statement** The study protocol, informed consent form, clinical study report and individual participant data that underlie the results reported in this article, after deidentification (text, tables, figures and appendices) will be made available upon request, immediately after publication and ending 4 years after

article publication. This information will be available to researchers who provide a methodologically sound proposal within the limits of the approval given by the original independent ethics review committee, or investigators whose proposed use of the data has been approved by another independent review committee ("learned intermediary") identified for this purpose, for individual participant data meta-analysis. Proposals should be directed to helen.maclaughlin@nhs.net. To gain access, data requestors will need to sign a data access agreement. Proposals may be submitted up to 36 months following article publication.

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
