## [Reviewer comments · BMJ Open]

This paper was submitted to a another journal from BMJ but declined for publication following peer review. The authors addressed the reviewers' comments and submitted the revised paper to BMJ Open. The paper was subsequently accepted for publication at BMJ Open.

(This paper received three reviews from its previous journal but only two reviewers agreed to published their review.)

ARTICLE DETAILS

TITLE (PROVISIONAL)	Obesity and recovery from acute kidney injury (Ob AKI): a prospective cohort feasibility study
AUTHORS	MacLaughlin, Helen L; Blacklock, Rochelle M; Wright, Kelly; Pot, Gerda; Jayawardene, Satish; McIntyre, CW; Macdougall, Iain; Selby, Nicholas M

VERSION 1 – REVIEW

REVIEWER	Mariko Miyazaki Associate Professor, Department of Nephrology, Endocrinology and Vascular medicine, Tohoku University Graduate School of Medicine; Director, Division of Blood Purification, Tohoku University Hospital, Japan
REVIEW RETURNED	13-Jun-2018

GENERAL COMMENTS	Thank you for giving the opportunity to review this article. This article mentioned no significant difference among various BMI. It was interesting for me. I wonder how about examining the possibility of "J shape" relationship between BMI and AKI risk. Because the person of a small BMI has frailty. Frailty may be the risk factor of AKI. Please consider further analysis if possible. Minor comments: The methodological comparison of GFR estimation is crucial in this article, I think some improvements will help to understanding easily the difference between creatinine based eGFR using MDRD, CKD-EPI, and cystatin C based eGFR. I suggest the use of abbreviations such as eGFRckd-epi, eGFRcys.
---

REVIEWER	Jill Vanmassenhove Ghent University Hospital, Belgium
REVIEW RETURNED	25-Jun-2018

GENERAL COMMENTS	Overall, this is a very interesting initiative. Authors studied the feasibility of a prospective observational study looking at the influence of obesity on AKI recovery and CKD post AKI. I have some comments: -Authors explain very well what the difficulties are when defining AKI in patients with obesity. I wonder how authors will be able to differentiate between a 'real' physiological difference in the response of obese vs non obese after an episode of AKI versus differences that are related to misinterpreting serum creatinine values, which might be more likely in the obese vs non obese group. Within
--

	individuals there might also be a bias if more severely ill obese vs non obese patients are included with higher fluctuations of weight between admission and discharge. In that regard, it might also be interesting to mention how many of these included patients are ICU patients. It seems appropriate to include patients that are not too heterogeneous and mostly differ in weight rather than in many other confounding variables/ non GFR determinants. -There is a lot of discussion on how to define recovery of kidney function and at what time point this should be done. It might also be difficult to distinguish between AKI still in recovery and CKD. And even in case of apparent recovery there is an increased risk of CKD development. Also, several factors such as severity of AKI, duration of AKI and presence of repetitive episodes of AKI will influence the risk of later CKD development, independent on presence of obesity, and should be taken into account. Presence of proteinuria will also increase the risk of CKD development, even if creatinine levels have returned to baseline. Recovery might be overestimated if there is a large change in body mass which is to be expected in severely ill ICU patients who might demonstrate a decrease in sCr value due to loss of muscle mass rather than to recovery of kidney function -The algorithm used for AKI definition and especially its limitations, should be better elaborated in the text. -I understand that authors only included patients who have a historical baseline sCr value available, however there is a reason why some patients have this value available and others don't which will inevitably introduce bias. The same can be said for AKI definition with some patients having had more sCr measurements than others during hospitalization. The more values there are, the higher the likelihood of detecting AKI in the first place. -Urinary output is not included in the AKI definition. Although the use of the urinary output criterion in the obese can definitely be problematic, a lot of cases might also be missed by not using this criterion. -Although the outcome 'change in CKD classification class' is a very practical one, it is rather mathematical and can be artificial e.g. if eGFR changes from 61 to 58 versus from 85 to 46, the change in CKD class will be the same, however these patients have a completely different trajectory.
--	--

VERSION 1 – AUTHOR RESPONSE

Reviewer Comment	Response	Page
Reviewer 1		
Examine the possibility of a J shaped relationship between BMI and AKI risk, and consider frailty as a risk factor. Consider further analysis if possible	Thank you for your observation that the relationship between BMI and AKI may be J shaped rather than linear. Whilst underweight is likely to be a risk factor for mortality, it is unknown if low BMI is associated with CKD outcomes after AKI. The study is not sufficiently powered for such an analysis. We acknowledge that frailty may be another risk factor for the development or progression of CKD after an episode of AKI. Frailty could also be considered a risk marker for co-morbidity and long-term chronic disease. Our study was a feasibility study with	17

	outcomes for recruitment and retention and contained exploratory outcomes related to obesity. There are no measures of frailty in our study and BMI is not a sufficient surrogate. We have acknowledged frailty as a potential confounder in the discussion. We have suggested that non-linear relationships are considered in future studies.	
The methodological comparison of GFR estimation is crucial in this article, I think some improvements will help to understanding easily the difference between creatinine based eGFR using MDRD, CKD-EPI, and cystatin C based eGFR. I suggest the use of abbreviations such as eGFR_{ckd-epi}, eGFR_{cys}	Thank you for your observation that methodological comparison of GFR estimation is crucial in examining this relationship. We have made clarifications in the text. Only the CKD-EPI equations were used in this study. CKD-EPI equations can be based on creatinine, or cystatin c, or a combination of the two. The MDRD equation was not used in this study. We have adjusted the text and used the following abbreviations: CKD-EPI cystatin c + creatinine equation - eGFR_{creat+cysC} CKD-EPI creatinine equation - eGFR_{creat} We did not use the MDRD equation or the CKD-EPI cystatin c based equation in this study.	throughout
Reviewer 2		
I wonder how authors will be able to differentiate between a 'real' physiological difference in the response of obese vs non obese after an episode of AKI versus differences that are related to misinterpreting serum creatinine values, which might be more likely in the obese vs non obese group.	We agree that interpretation of serum creatinine values with obesity are difficult and have attempted to address this in our study by acknowledging differences in eGFR using different estimating equations. Our examination of the literature indicated that the use of both creatinine and cystatin C in the estimating equation improves the estimation of GFR. We have used the CKD-EPI creatinine + cystatin C equations for all time points after AKI, as cystatin C is not routinely measured. These problems are exacerbated when weight changes significantly between measurements, particularly when weight decreases, and the error associated with larger body size decreases at the same time. In our study, body weight increased in all groups over the study period, an expected occurrence after a period of acute illness. Furthermore, we expect that defining CKD development or progression as at least a 25% drop in eGFR together with a decline in CKD stage, would reflect a "real" physiological difference, rather than only variation in an estimating equation, particularly when intra-individual differences	

	are calculated.	
Within individuals there might also be a bias if more severely ill obese vs non obese patients are included with higher fluctuations of weight between admission and discharge. In that regard, it might also be interesting to mention how many of these included patients are ICU patients.	Thank you for raising this important point. Indeed, absolute changes in weight may be higher in obese vs non obese patients who are severely ill. However, typically the percentage weight loss is similar, and this is why definitions of acute malnutrition use percentage weight loss rather than absolute changes in weight to assess risk. We assessed the severity of the AKI but we did not record whether the admission included an ICU stay or not. We agree that this would be an interesting variable to include in an analysis of the effect of BMI on CKD risk after AKI in a future study.	
There is a lot of discussion on how to define recovery of kidney function and at what time point this should be done. It might also be difficult to distinguish between AKI still in recovery and CKD. And even in case of apparent recovery there is an increased risk of CKD development. Also, several factors such as severity of AKI, duration of AKI and presence of repetitive episodes of AKI will influence the risk of later CKD development, independent on presence of obesity, and should be taken into account.	Thank you for your insightful comments. We agree that the recovery of kidney function is difficult to define and measure. Differences in recovery at 6 months and the development or progression of CKD at 12 months in the present study highlight that recovery or otherwise at 6 months may not be associated with later outcomes. We completely agree that severity of AKI, duration of AKI, and repetitive AKI episodes will influence the risk of later CKD development. These outcomes were measured in our study, although not reported in this manuscript. We agree that these variables should be taken into account in a study sufficiently large enough to examine risk factors for CKD development and progression after AKI. As our study was a feasibility study on 100 patients, we are unable to adequately examine the impact of these factors on outcomes. We have acknowledged that these factors should be taken into account in our discussion.	17
Presence of proteinuria will also increase the risk of CKD development, even if creatinine levels have returned to baseline. Recovery might be overestimated if there is a large change in body mass which is to be expected in severely ill ICU patients who might demonstrate a decrease in sCr value due to loss of muscle mass rather than to recovery of kidney function	We agree that the serum creatinine may decrease due to a loss of muscle mass after severe illness, and this will impact upon the eGFR. This would be expected in both obese and non-obese patients in severe illness. We have included cystatin-C in our eGFR estimating equation to overcome some of the limitations of using creatinine based equations. As stated above, the short-term recovery from AKI may not necessarily affect longer-term outcomes related to CKD development and progression. We agree that proteinuria is a likely confounder in the relationship between AKI	17

	and CKD. Proteinuria was measured at all timepoints in the current study and it should be included in multivariable analyses in a sufficiently large study. It is beyond the scope of the current study to classify patients by proteinuria status in outcome analyses.	
The algorithm used for AKI definition and especially its limitations, should be better elaborated in the text.	We have included a section on the limitations of the KDIGO definition and the use of an algorithm for AKI detection in the discussion.	17
I understand that authors only included patients who have a historical baseline sCr value available, however there is a reason why some patients have this value available and others don't which will inevitably introduce bias. The same can be said for AKI definition with some patients having had more sCr measurements than others during hospitalization. The more values there are, the higher the likelihood of detecting AKI in the first place.	We acknowledge that availability of baseline serum creatinine values and the number of creatinine measurements during hospitalisation does introduce bias. However, this is a limitation of all studies on AKI and is not limited to the present study.	17
Urinary output is not included in the AKI definition. Although the use of the urinary output criterion in the obese can definitively be problematic, a lot of cases might also be missed by not using this criterion.	Urinary output is included in the KDIGO definition of AKI used in the current study. We have clarified this for the identification of potential participants from referrals to the nephrology service. With the following phase added "using the full KDIGO definition of AKI". We acknowledge that the urinary criterion is omitted in the AKI detection algorithm used in this study. This is common to many studies on AKI as urinary output is not systematically measured or recorded for all hospitalised patients in a reliable way.	5 17
Although the outcome 'change in CKD classification class' is a very practical one, it is rather mathematical and can be artificial e.g. if eGFR changes from 61 to 58 versus from 85 to 46, the change in CKD class will be the same, however these patients have a completely different trajectory.	The definition of progression includes both a 25% reduction in eGFR PLUS a decline of at least 1 category in CKD staging within the same patient. Therefore a change in eGFR from 61 to 58 would not meet the full criteria for CKD development or progression. We also looked at the frequency of experiencing a 25% decline in eGFR without a change in CKD category, which could lead to an underestimation of progression, however this occurred in 1 participant only.	7

VERSION 2 – REVIEW

REVIEWER	M Miyazaki Tohoku University Graduate School of Medicine, Japan
REVIEW RETURNED	18-Sep-2018

GENERAL COMMENTS	This study deals with the relationship between obesity and recovery or progression of kidney function after acute kidney injury. I agree that chronic kidney disease after AKI is important issue, however, the results in this study showed several difficulties to demonstrate clearly the effects of obesity as previous reviewers' comments. I would like to update study design and examination factor based on this feasibility research data, and wait authors submit again in future.
---

REVIEWER	Jill Vanmassenhove Ghent University Hospital Belgium
REVIEW RETURNED	03-Sep-2018

GENERAL COMMENTS	The authors addressed all of my concerns. I have no further comments.
---